

# Heteronuclear and Homonuclear Radio Frequency Driven
# Recoupling
**Authors:** Evgeny Nimerovsky*, Kai Xue, Kumar Tekwani Movellan & Loren B. Andreas*
**Affiliations:**
Department of NMR based Structural Biology, Max Planck Institute for Biophysical Chemistry, Am
Fassberg 11, Göttingen, Germany
*Corresponding authors: land@nmr.mpibpc.mpg.de   ORCID:   0000-0003-3216-9065   and
evni@nmr.mpibpc.mpg.de
**Abstract**
The Radio Frequency Driven Recoupling (RFDR) pulse sequence is used in magic-angle spinning
(MAS) NMR to recouple homonuclear dipolar interactions. Here we show simultaneous recoupling of both
the heteronuclear and homonuclear dipolar interactions by applying RFDR pulses on two channels. We
demonstrate the method, called HETeronuclear RFDR (HET-RFDR) on microcrystalline SH3 samples at
10 kHz and 55.555 kHz MAS. Numerical simulations of both HET-RFDR and standard RFDR sequences
allow better understanding of the influence of offsets, paths of magnetization transfers for both HET-RFDR
and RFDR experiments as well as the crucial role of XY phase cycling.
Keywords: Magic Angle Spinning NMR, heteronuclear and homonuclear RFDR, the operator analysis
**Introduction**
Magic-angle spinning (MAS) NMR spectroscopy is used to obtain atomic resolution spectra of
materials and biological molecules in the solid state, by removal of the broadening associated with
anisotropic dipolar couplings and other interactions. Under control of radio frequency pulses, dipolar



interactions can be switched on, or recoupled, in order to correlate nearby spins or to accurately determine
internuclear distances. Recoupling sequences can be broadly categorized as homonuclear (Bennett et al.,
1992; Ok et al., 1992; Zhang et al., 2020; Gelenter et al., 2020; Takegoshi et al., 2001; Szeverenyi et al.,
1982; Hou et al., 2011b, 2013; Carravetta et al., 2000; Bennett et al., 1998; Nielsen et al., 2012) or
heteronuclear (Gelenter et al., 2020; Gullion and Schaefer, 1989; Jaroniec et al., 2002; Hing et al., 1992;
Hartmann and Hahn, 1962; Rovnyak, 2008; Metz et al., 1994; Hediger et al., 1994; Hou et al., 2011a;
Brinkmann and Levitt, 2001; Gelenter and Hong, 2018; Zhang et al., 2016; Nielsen et al., 2012). The
homonuclear Radio Frequency Driven Recoupling (RFDR) sequence (Bennett et al., 1992)  has been
successfully applied for the qualitative and quantitative determinations of the dipolar spin correlations in
materials (Saalwächter, 2013; Messinger et al., 2015; Fritz et al., 2019; Roos et al., 2018; Nishiyama et al.,
2014a; Wong et al., 2020; Hellwagner et al., 2018; Pandey and Nishiyama, 2018) and biomolecular samples
(Zheng et al., 2007; Tang et al., 2011; Shen et al., 2012; Pandey et al., 2014; Grohe et al., 2019; Andreas et
al., 2015; Petkova et al., 2002; Aucoin et al., 2009; Zinke et al., 2018; Zhang et al., 2017; Zhou et al., 2012;
Jain et al., 2017; Colvin et al., 2015; Shi et al., 2015; Daskalov et al., 2020). Sun et al. (1995) showed that
the RFDR pulse sequence element could also be used as a part of the SPICP experiment (Wu and Zilm,
1993) for removing the undesired effect of the chemical shift terms to zero order.

38        Depending on the assumptions (Ok et al., 1992; Ishii, 2001), two different Average Hamiltonian

Theory (Haeberlen and Waugh, 1968; Maricq, 1982) (AHT) descriptions have been detailed for RFDR. In
both, homonuclear dipolar recoupling occurs via a rotor-synchronized train of $\pi$-pulses, with one pulse each
rotor period (Bennett et al., 1992) on a single channel. In the first case, delta $\pi$-pulses are assumed (Bennett
et al., 1992). The efficiency of recoupling is linked with the rotational resonance conditions (Bennett et al.,
1992, 1998), and depends on the ratio between chemical shift offset difference and MAS rate. In the second
theoretical description, the effects of finite $\pi$-pulses are considered (Bennett et al., 1992; Ishii, 2001;
Nishiyama et al., 2014b; Zhang et al., 2015; Brinkmann et al., 2002; Ji et al., 2020). The efficiency of
recoupling in this case depends on a duty factor (Ishii, 2001), defined as the fraction of the rotor period

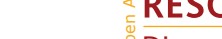

occupied by the π-pulse. The RFDR pulses are applied according to a variety of xy phase cycling schemes,
which have been analyzed with the intent to suppress imperfections associated with offset differences, rf-
field inhomogeneity and second order Average Hamiltonian terms between different anisotropic
interactions (Zhang et al., 2015).
The full high field truncated dipolar Hamiltonian of the homonuclear $I_2$ spin system is represented
as follows:
$$H_{D,Full}^{II} = \omega_{D,12}(t)[3I_{z1}I_{z2} - \bar{I}_1 \cdot \bar{I}_2]. \quad \text{Eq. (1)}$$
where $\omega_{D,12}(t)$ is a periodic time dependent function (Olejniczak et al., 1984) that depends on the
positions of spins $I_1$ and $I_2$ within the rotor. This Hamiltonian is subsequently referred to as the full
Hamiltonian, and contains only the A and B terms of the dipolar alphabet (Slichter, 1990).
The interesting conclusion can be obtained, if we simplify the Eq. (1). The dipolar Hamiltonian
during RFDR can be simplified (in the absence of other interactions) by considering that $\bar{I}_1 \cdot \bar{I}_2$ commutes
with the secular part ($I_{z1}I_{z2}$) and with the rf-field Hamiltonian. At the end of each rotor period, the
oscillatory $\omega_{D,12}(t)$ term ensures zero total evolution. The simplified Eq. (1) is:
$$H_{D,M}^{II} = 1.5\omega_{D,12}(t)2I_{z1}I_{z2}. \ \text{Eq. (2)}$$
Comparing Eq. (2) with full Dipolar Hamiltonian of the heteronuclear *IS* spin system(Mehring, 1983):
$$H_{D,Full}^{IS} = \omega_{D,12}(t)2I_zS_z, \ \text{Eq. (3)}$$
we notice that the difference between Eq. (3) and Eq. (1) is a factor of 1.5. Note that we have made the
substitution of $I_{z1}$ to $I_z$ and $I_{z2}$ to $S_z$ while the dipolar function, $\omega_{D,12}(t)$, has been kept the same. Such
comparison suggests a HETeronuclear-RFDR (HET-RFDR), which should have a scaling of 1.5 as
compared with the homonuclear case.



68   In this article we investigate spin dynamics under HET-RFDR, in which RFDR π-pulses are

69  applied simultaneously on two channels (Figure 1). We demonstrate simultaneous heteronuclear and

70  homonuclear transfers using HET-RFDR applied to α-PET (Movellan et al., 2019) labeled SH3 at 10 kHz

71  and 55.555 kHz MAS.

72   We perform and compare a numerical operator analysis of both RFDR and HET-RFDR

73  experiments under different simulated conditions. This numerical analysis allows to define the conditions

74  under which homonuclear and heteronuclear RFDR polarization transfers have similar behaviors, to

75  understand the paths through which the signals are transferred between operators, and to understand the

76  crucial role of 90 degree phase alternation (XY-4, XY-8, etc) (Ishii, 2001; Nishiyama et al., 2014b; Zhang

77  et al., 2015; Hellwagner et al., 2018) for both RFDR and HET-RFDR recoupling.

78  **HET-RFDR Experiments**

79   Figure 1 shows two 2D (H)N(H)H pulse sequences used to evaluate the HET-RFDR transfer. For

80  both sequences, the transfer from proton to nitrogen is implemented with ramped cross polarization (CP)

81  and then the nitorgen dimension is encoded ($t_1$) for 2D spectra,).In Figure 1A, the transfer to structurally

82  interesting protons is implemented with N to H CP followed by H-H RFDR. In Figure 1B, the same

83  transfer is implemented with a single HET-RDFR period. The HET-RFDR transfer avoids the back CP

84  step. Instead, nitrogen polarization is placed along the $\hat{z}$ axis and transfered to directly bonded proton

85  spins and at the same time to remote proton spins with the simultaneous application of the π-pulses on the

86  proton and nitrogen channels.

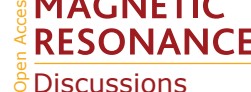

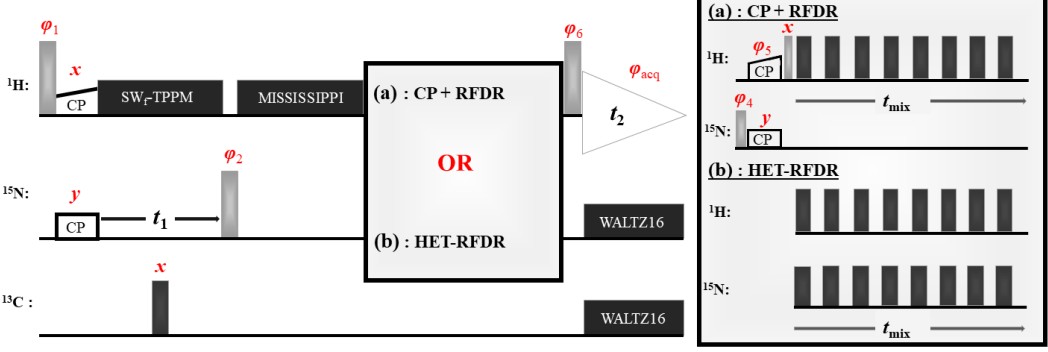

87

**Figure 1** Two versions of the (H)N(H)H pulse sequence are shown. The first, (a), is the standard implementation with CP +

RFDR. The second, (b), instead uses the new HET-RFDR recoupling element. Light grey pulses represent π/2-pulses, whereas

dark grey pulses represent π-pulses. The ramped CP transfer from proton to nitrogen as well as from nitrogen to proton in (a) are

indicated with constant power on the nitrogen channel and a ramp in power on the proton chanenl. During the inderect dimension

($t_1$), SW$_f$-TPPM decoupling is applied at 55 kHz , repectively. A single π-pulse in the middle of $t_1$ decouples carbon. Water

supression is implemented with the MISSISSIPI (Zhou and Rienstra, 2008) sequence. During acquisition, WALTZ16 (Thakur et

al., 2006) decoupling is applied on nitrogen and carbon channels. The phases are: $\varphi_1 = x, -x$; $\varphi_{acq} = y, -y, -y, y, -y, y, y, -y$.

In (a) the phases are: $\varphi_2 = x$; $\varphi_4 = x, x, -x, -x$; $\varphi_5 = y, y, y, y, -y, -y, -y, -y$; $\varphi_6 = x$. In (b) the phases are: $\varphi_2 = x, x, -x, -x$; $\varphi_6 = x, x, x, x, -x, -x, -x, -x$. RFDR π-pulses on both channels use the XY8 scheme (Gullion et al., 1990).

97        Figure 2 compares the 1D and 2D spectra obtained with the two sequences of Figure 1. In Figure

2a, the 1D signal is shown as a function of RFDR mixing time. For the standard sequence (blue) the N to
H CP was 0.55 ms. The HET-RFDR signal is shown in (red). For the directly bonded amide protons, the
HET-RFDR polarization transfer achieves only ~40% of the CP signal. This occurs at 0.846 ms mixing
(second red spectrum). However, with increased mixing of about 3 ms, HET-RFDR reaches the same
efficiency as the standard sequence. This is notable since transfer over long distances has been
implemented with ~3 ms mixing for deuterated samples (Grohe et al., 2019; Linser et al., 2014).

104        Structurally interesting cross-peaks are indeed observed in the 2D HET-RFDR spectrum shown in

Figure 2b at both 1  ms (red) and 3 ms (cyan) mixing. For comparison, the (H)NH spectrum (without
RFDR mixing) is shown in black, to indicate the directly bonded amide correlations. For example, we
have observe the amide-amide contact between V44 and V53, which is 4.82 Å in the crystal (pdb code



2NUZ (Castellani et al., 2002)). The amide to side chain contact of a A55 N to Hβ (3.41 Å) is also
indicated in the Figure, along with a sequential contact from Y13 $^{15}$N to L12 $^{1}$Hα, which is 3.26 Å. These
peaks are boxed in Figure 2b, and the 1D slices shown above the 2D spectra. Comparing cross peak
intensities with the two mixing times, we observe an increase in signal for the longer 3 ms mixing,
indicating that the HET-RFDR sequence is indeed competitive for recording structurally interesting
contacts.

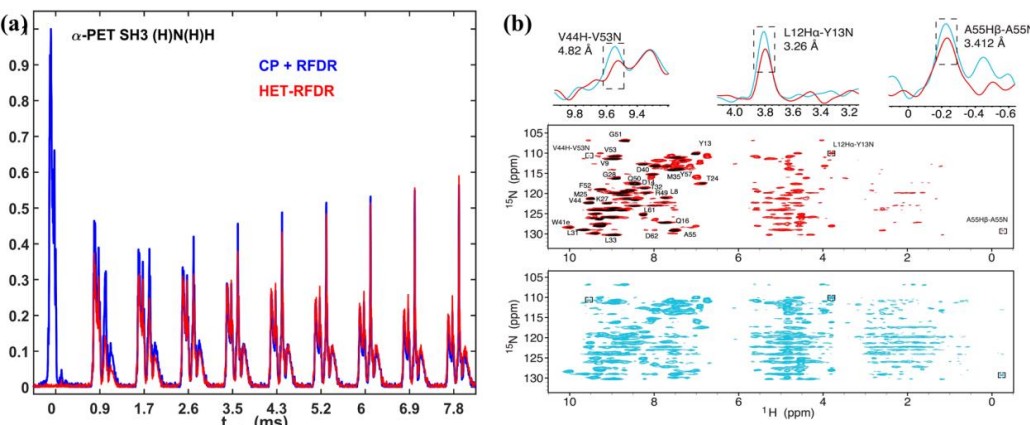


**Figure 2** 1D (a) and 2D (b) (H)N(H)H spectra of α-PET labeled SH3. For all spectra the first CP from proton to nitrogen was
performed with 1.05 ms. (a) 1D spectra with different sequences used for the second transfer: CP + RFDR (blue) and HET-RFDR
(red). For CP + RFDR, 0.55 ms of CP was used. For both RFDR and HET-RFDR, $t_{mix}$ of 0, 0.846, 1.728, 2.592, 3.456, 4.32,
5.184, 6.048, 6.912, 7.7776 msare shown. (b) 2D HET-RFDR at two mixing times: 1.152 (red) and 3.456 (blue) ms. For
comparison, the CP based (H)N(H) spectrum is overlayed in black showing in-residue correlations. Spectra were recorded at a
600 MHz Bruker instrument equiped with a  1.3 mm probe and an MAS frequency of 55 kHz. The widths of π-pulses on proton
and nitrogen channels were 5.8 us and 6.6 us, respectively. The experimental parameters are detailed in Table 1 and 2 the
'Experimental Methods'.
At 55 kHz MAS on a 600 MHz instrument, the chemical shift offsets can always be much smaller
than the spinning frequency. At a lower MAS frequency, the offsets become important for HET-RFDR.
The recoupling then depends on a heteronuclear 'offset difference' that we define as $\Delta\Omega_{ij} = \Omega_i - \Omega_j$,
where $\Omega_i$ and $\Omega_j$ are the offsets on each channel (the difference between the Larmor frequency of the spin



**MAGNETIC RESONANCE**
Open Access Discussions

and the carrier frequency (Bak et al., 2000)). The explanation of the following behavior is described in the
"Numerical Operator Analysis" section. When $\Omega_i = \Omega_j = 0$ as well as $\Delta\Omega_{ij} = \Omega_i - \Omega_j \approx n\nu_R$ (n=0, ±1,
±2…), the HET-RFDR polarization transfer reaches local maximal intensities. However, when $\Delta\Omega_{ij} =$
$\Omega_i - \Omega_j \approx 0.5n\nu_R$ (n=±1, ±3…), the HET-RFDR polarization transfer reaches local minima. The
experimental confirmation of this is shown in Figure 3, where the effect of different proton and carbon
offsets is explored for proton-carbon HET-RFDR spectra. The spinning frequency was reduced to 10 kHz
MAS for these measurements and the signal detected on the carbon channel. The 1D HC HET-RFDR
pulse sequence is shown in the SI (Figure S1).
Figures 3a-e depicts the HET-RFDR spectra when the carbon carrier frequency is changed (numbers
show the offset from the alpha carbon at ~53 ppm), whereas the alpha proton offset is kept at 0 kHz (at
4.6 ppm). While heteronuclear transfer is detected at zero offset (Figure 3a) or with 11.1 kHz carbon
offset (Figure 3e), the signal remains in the noise when the carbon offset is 5.85 kHz (Figure 3c).
A similar effect can be detected when the proton carrier frequency is changed (increased from 4.6 ppm),
but this time the carbon offset is set to 5 kHz from Cα (83.66 ppm) to show that it is the offsets on both
channels ($\Delta\Omega_{C\alpha H\alpha}$) that is important (Figures 3f-j). The series of spectra show a local minimal transfers at
offset differences of 5 kHz (Figure 3f) and -5 kHz (Figure 3h) and local maximal polarization transfers at
differences of 0 (Figure 3g) and -10 kHz (Figure 3j).




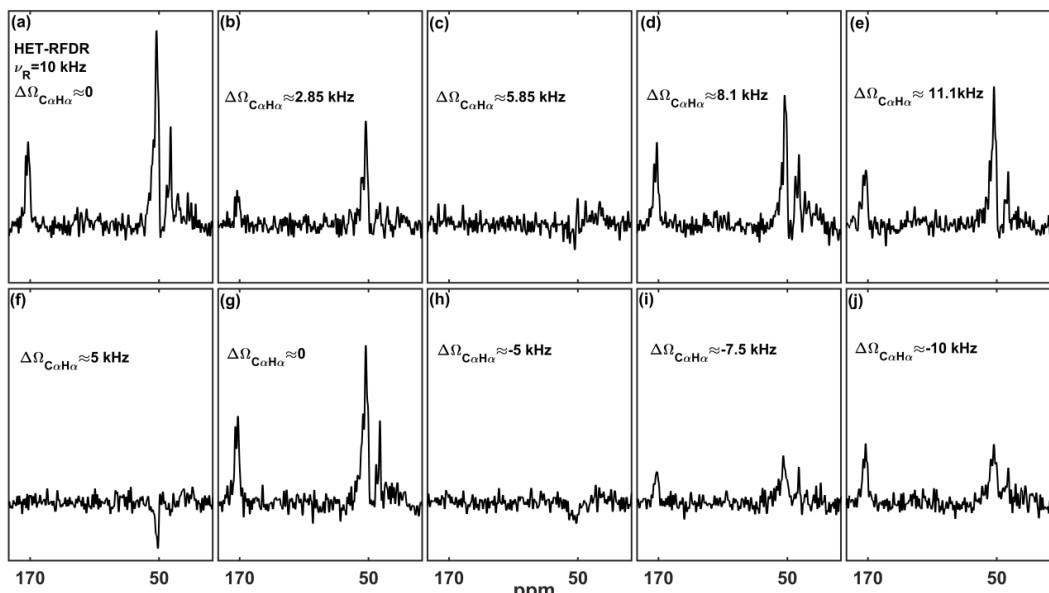

**Figure 3** The influence of the carbon and proton offsets on proton-carbon HET-RFDR polarization transfers at 4.8 ms mixing.
α-PET labeled SH3 was used with 10 kHz MAS at a 600 MHz spectrometer using a 1.3 mm probe. The widths of π-pulses on
proton and carbon channels were 5.8 us and 6.6 us, respectively. For (a)-(e) the proton carrirer frequency was set to 4.6 ppm and
carbon carrier frequency was set to 51 ppm (a), 70 ppm (b), 90 ppm (c), 105 ppm (d) 125 ppm (e). For (f)-(j) the carbon carrirer
frequency was set to 83.66 ppm and the proton carrier frequency was set to 4.6 ppm (f), 12.933 ppm (g), 21.26 ppm (h), 25.43
ppm (i) and 29.6 ppm (j). The indicated offset differences, $\Delta\Omega_{C\alpha H\alpha} = \Omega_{C\alpha} - \Omega_{H\alpha}$ in kHz were calculated based on typical
isotropic chemical shifts of $C_\alpha$ (51 ppm) and $H_\alpha$ (4.6 ppm) at a 600 MHz spectrometer. The experimental parameters are detailed
in Table 1 and 2 the 'Experimental Methods'. The 1D HET-RFDR sequence is shown in the SI (Figure S1).

### Numerical Operator Analysis

154         To comprehend the mechanism underlying the transfers during the HET-RFDR and also the well-

known RFDR pulse sequence, we use a numerical simulation approach. We identify the conditions under
which the heteronuclear and homonuclear spin systems under HET-RFDR and RFDR sequences have
similar behaviors. Considering the evolutions of the different spin systems through HET-RFDR and
RFDR during the first two rotor periods, we identify the operators that are involved in the polarization
transfer.

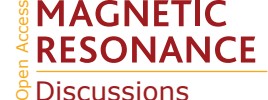

To identify the conditions under which the HET-RFDR and RFDR sequences have similar and
different behaviors we simulated a three spin system at high (55.555 kHz) and low (10 kHz) MAS
frequencies. In Figure 4 we compare the RFDR transferred signals for $I_3$ (a homonuclear 3-spin system,
black lines) and HET-RFDR transferred signals for $ISR$ (three different types of spins with the names $I$, $S$
and $R$; red lines) spin systems. At 55.555 Hz MAS when the offset difference is small compared to MAS
rate, the behavior of the homonuclear $I_3$ spin system is similar to the behavior of the heteronuclear $ISR$
spin system (Figure 4a). However, when the MAS rate is low (10 kHz) and the offset difference cannot be
neglected, the behaviors of these spin systems are completely different (Figure 4b). For the homonuclear
spin system ($I_3$), the polarization transfers are efficient for all dipolar pairs (black lines), whereas for the
heteronuclear spin system ($ISR$) the HET-RFDR polarization transfer is detected between $R$ and $I$ spins
(Figure 4b, red dashed-dotted line) only. For this $RI$ pair the offset difference was chosen as 10 kHz,
whereas for the other spin pairs ($SI$, $RS$) the offset differences were set to 5 kHz. These simulations show
a special condition of $\sim 0.5\nu_R$ of offset difference for the heteronuclear spins under which the transfer is
negligible. The simulations are in full agreement with the experiments, which were shown in Figure 3.
Another interesting observation can be made from the influence of the offset difference on the RFDR
transfer for the homonuclear $I_3$ spin system (Figure 4b, black lines). For a 5 kHz of offset difference, the
RFDR polarization transfer between $I_{z2}$ and $I_{z3}$ spins is significantly faster with 10 kHz MAS (Figure 4b,
black dashed line) than at 55.555 kHz MAS (Figure 4a, black dashed line). Since the duty factor is
decreased with decreasing MAS frequency(Ishii, 2001): 0.33 for 55.555 kHz MAS and 0.06 for 10 kHz
MAS, the opposite behavior is expected if one considers only the effect of finite pulses in the RFDR
experiment(Ishii, 2001). It indicates that when the offset difference cannot be neglected with respect to
the MAS rate, it has a significant influence on the RFDR transfer efficiency between homonuclear spins
despite the significant remoteness from the rotational resonance condition (Bennett et al., 1992, 1998).

**MAGNETIC RESONANCE**
Open Access Discussions

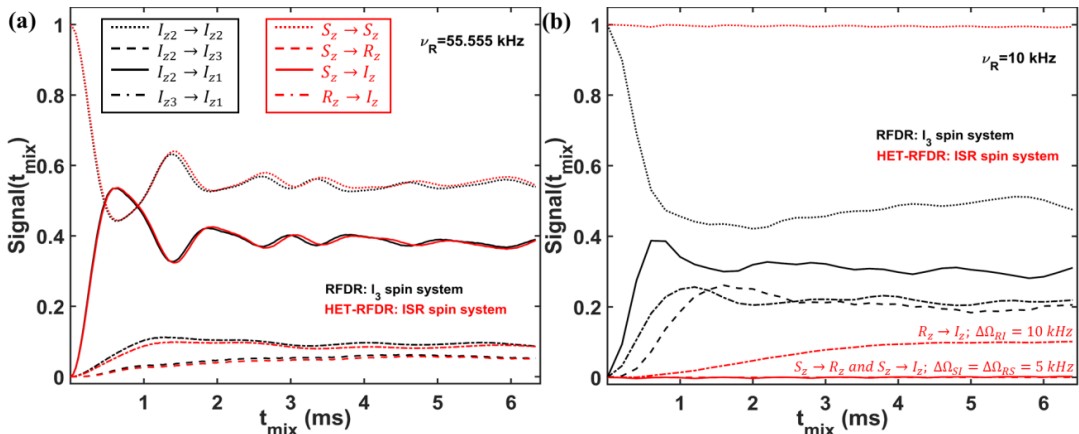


**Figure 4** Comparison of the simulated RFDR and HET-RFDR signals. $I_3$ (three homonuclear spins, black lines) and $ISR$ (three
different spin types, red lines) for 55.555 kHz (a) and 10 kHz (b) MAS. 83 kHz of rf-field is used (6 us of the widths of π-pulses).
The vertical axis shows the intensities of the starting and transferred signals between different operators of $I_3$ and $ISR$ spin
systems, respectively (the initial operator → the measured operator): $I_{z2} \to I_{z2}$ and $S_z \to S_z$ – (the dotted lines); $I_{z2} \to I_{z3}$ and
$S_z \to R_z$ – (the dashed lines); $I_{z2} \to I_{z1}$ and $S_z \to I_z$ – (the solid lines); $I_{z3} \to I_{z1}$ and $R_z \to I_z$ – (the dashed-dotted lines). For
both spin systems the offset (Ω) and CSA values are: [-3; 2; 7] (kHz) and [5.2; 2.5; 3].The dipolar coupling constants for
homonuclear spin system ($I_3$) spin system are: $\nu_{12,D} = 7.333$ kHz, $\nu_{13,D} = 2$ kHz, $\nu_{23,D} = 0.333$ kHz. For $ISR$ spin system all
dipolar constants are 1.5 times larger: $\nu_{IS,D} = 11$ kHz, $\nu_{IR,D} = 3$ kHz, $\nu_{SR,D} = 0.5$ kHz. The simulated measurements occurs
every 2 rotor periods. XY8 phase cycling is used. $I_{z1} \to I_{z1}, I_{z3} \to I_{z3}, I_z \to I_z$ and $R_z \to R_z$ are not shown.

193       In order to understand the different dependence on offset difference for the heteronuclear and

homonuclear spin systems as well as via which operators the polarization transfer occurs, we considered
the evolutions of two systems - $I_2$ homonuclear and $IS$ heteronuclear spin systems - under RFDR and
HET-RFDR sequences with 10 kHz MAS. We simulated the polarization transfers between different
operators during the first two rotor periods, which completes the basic RFDR element: $t(\pi_x) \to del_1 \to$
$t(\pi_y) \to del_2$. We consider the amplitudes of the operators for a single molecular orientation since it
allows to see the significant evolution of the operators during the two rotor periods. Figure 5a,c,e shows
the amplitudes of four Cartesian operators (Ernst et al., 1987) for $IS$ (HET-RFDR) and Figures 5b,d,f





shows the operators for $I_2$ (RFDR) spin systems. The measured Cartesian operators are $I_z, S_z, 2I_xS_y, 2I_yS_x$
and $I_{z1}, I_{z2}, 2I_{x1}I_{y2}, 2I_{y1}I_{x2}$ for $IS$ and $I_2$ spin systems, respectively.
The evolutions of four operators *during* two rotor periods for the $IS$ spin system the $I_2$ spin system are
different, regardless of the offset difference. However, with a zero offset difference, the simulated
heteronuclear operators (Figure 5a) and the homonuclear operators (Figure 5b) show the same values of
the amplitudes at one and two rotor periods. From the 64 possibilites (details in the SI, section 'The
Operator Paths') for magnetization tranfer between heteronuclear operators $I_z$ and $S_z$ during the two rotor
periods, we find only one path with nonzero amplitude: $I_z \xrightarrow{\pi_x} 2I_xS_y \xrightarrow{del_1} 2I_xS_y \xrightarrow{\pi_y} S_z \xrightarrow{del_2} S_z$. In contrast to
the single path found for HET-RFDR, for the homonuclear case all 64 paths connecting operators $I_{z1}$ and
$I_{z2}$ have non-zero amplitudes. However, after each rotor period, the sum of all homonuclear paths provides
the same values of the amlitudes as for the heteronuclear $IS$ spin system.
In contrast, with a non-zero offset difference, the amplitudes of homonuclear and heteronuclear operators
do not coinside at any time (Figures 5c and d). Moreover, while the amplitude of $I_{z1} \rightarrow I_{z2}$ polarization
transfer is sigificantly increased (Figure 5d, green line), the corresponding heteronuclear amplitude for
$I_z \rightarrow S_z$ transfer is sigificantly decreased (Figure 5c, green line).
In Figure 3 we experimentally demonstrated (it also was shown with simulations in Figure 4b), that when
the offset differences, $\Delta\Omega_{C\alpha H\alpha}$, were not neglected with respect to MAS rate, the minimal and maximal
HET-RFDR polarization transfers were achieved under $\pm\sim0.5\nu_R$ and $0, \pm\sim\nu_R$, respectively. Such
dependence on the offset difference for the heteronuclear spin system can be understood from Figure 5c,
where the simulations are performed with $0.5\nu_R$ offset difference. During the first $\pi_x$ pulse the starting
signal is transferred from $I_z$ to $2I_xS_y$. Because of the offset difference of $0.5\nu_R$, the amplitude of this
operator is mainly transferred to $2I_yS_x$ during the first delay (Figure 5c, red line). Since the second $\pi$-pulse
has phase y, there is no transfer from $2I_yS_x$ to $I_{z2}$ and very little $I_z \rightarrow S_z$ polarization transfer overall by
the end of the second rotor period (Figure 5c, green line). Under $\pm\sim\nu_R$ offset difference the operator $2I_xS_y$



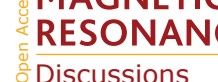

is transferred to itself during that first rotor period. Therefore the local maximal HET-RFDR polarization
transfer is detected. The sign, "~", is used since we consider the spin evolution with finite pulses.
Such dependence of the *IS* spin system on the offset difference indicates the importance of the
phase cycling for RFDR and HET-RFDR sequences. Figures 5d and f show the evolution of the operators
when there is no offset and both π-pulses have the same phase cycling – XX. For *IS* spin system (Figure
5e) only two operators have nonzero amplitudes during the investigated time: $I_z$ (black line) and $2I_xS_y$
(blue line), whereas $S_z$ and $2I_yS_x$ are not created. For the $I_2$ spin system (Figure 5d) all four operators
envolve during these two rotor periods. However, by the end of two rotor periods only two operators have
nonzero amplitudes, as for the *IS* spin system. In neither case is there magnetization transfer from $I_z$ to $S_z$
nor from $I_{z1}$ to $I_{z2}$ after one or two rotor periods. The formal proof of zero transfer signal for homonuclear
two spin system in the absence of offset difference can be found in the SI, "RFDR Phase Cycling"
section.
Additional spectra and simulation results are found in the supporting information. We recorded
proton-carbon HET-RFDR spectra using fully protonated [$^{13}$C, $^{15}$N] labeled SH3. We numerically
simulated multi-spin systems, either containing two protons and two carbons, or one nitrogen and two
protons, in order to track more complex transfer of magnetization. The main conclusions from the
simulations and the experiments in the SI are the agreement between experimental and simulated HET-
RFDR, and the expected small dependence of the HET-RFDR recoupling on the flip angle deviations
with XY8 phase cycling (Gullion et al., 1990).





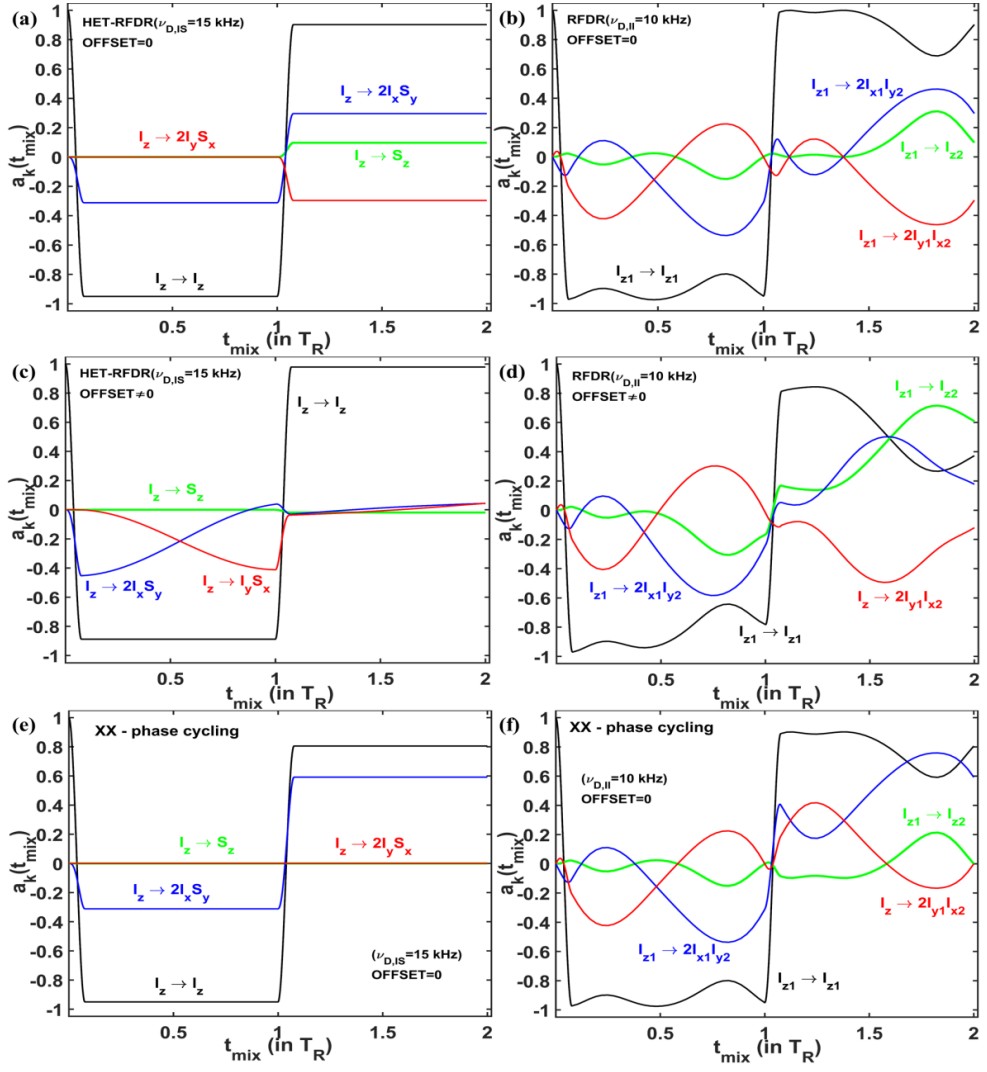


**Figure 5** The operator evolution through HET-RFDR and RFDR over two rotor periods. The simulated amplitudes of the operators of a single crystal (Euler angles: 184°; 141°; 349°) for HET-RFDR ((a), (c)) and RFDR ((b), (d)). For the heteronuclear $IS$ spin system, ($\nu_{D,IS} = 15$ kHz, the initial operator is $I_z$) and for the homonuclear $I_2$ spin system, ($\nu_{D,II} = 10$ kHz, the initial operator is $I_{z1}$). The MAS frequency was 10 kHz and the rf-field was 65 kHz. Black lines – $I_z$ and $I_{z1}$; Green lines – $S_z$ and $I_{z2}$; Blue lines – $2I_xS_y$ and $2I_{x1}I_{y2}$; Red lines – $2I_yS_x$ and $2I_{y1}I_{x2}$. For (a) – (d) the phases of the first and second $\pi$-pulses are X and Y, respectively. (e) and (f) show the case of $IS$ and $I_2$ spin systems, respectively, when the phases of the first and second $\pi$-pulses are both X. (a), (b), (e), (f) – Offset values in kHz: 0, 0. (c) and (d) – Offset values in kHz: 2, -3.

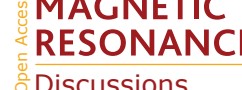

## Conclusion

In this article we firstly demonstrated HETeronuclear RFDR recoupling, when π-pulses with XY8 phase cycling were applied simultaneously on two channels. Observation of simultaneous heteronuclear and homonuclear polarization transfers as well as long range contacts were observed in 2D (H)NH spectra using HET-RFDR for the microcrystalline protein SH3 using α-PET labeling. The comparison of 1D HET-RFDR with CP followed by homonuclear RFDR showed similar efficiency of both methods at long mixing times of about 3ms and longer. We experimentally demonstrated and numerically explained the dependence of the HET-RFDR efficiency on the offset difference between dipolar coupled spins. A numerical operator analysis of both HET-RFDR and RFDR sequences showed that when the offest difference was small with respect to the MAS frequency, and with measurement at a whole number of rotor periods, the behavior of HET-RFDR was similar to the well-known homonuclear RFRD. However, different behaviors were observed when the offset difference could not be neglected.

Considering the evoultion of a single crystal during HET-RFDR and RFDR, we showed the operators that were responsible for the transfer. We demostrated that XY phase cycling of π-pulses has a crucial role for both HET-RFDR and RFDR transfer. With phase cycling of XX (or X$\overline{\text{X}}$) the transfers between heteronuclear and homonuclear spins did not occur in the absence of offsets. With the presence of the offset differences when they cannot be neglected in comparison to the MAS rate, RFDR polarization transfer with phase cycling of XX or X$\overline{\text{X}}$ does occur, although with lower efficiency as was described before (Bennett et al., 1992).

## Experimental methods

*Sample preparation*: Microcrystalline chicken alpha spectrin SH3 protein was used for acquisition of all experimental data. The samples were labeled with 100% protonation at exchangeable sites and either with alpha proton exchange by transamination (α-PET) or with uniform $^{13}$C and $^{15}$N labeling with the protocol described in (Movellan et al., 2019).



*Simulations*: HET-RFDR and RFDR simulations were performed with in-house MATLAB scripts using
numerical solution of the equation of motion (Nimerovsky and Goldbourt, 2012).
*Solid state NMR spectroscopy*: The HC and (H)N(H)H spectra of α-PET SH3 were acquired at 14.1 T (600
MHz) using a Bruker AVIIIHD spectrometer using a MASDVT600W2 BL1.3 HXY probe. The
experiments were performed at 10 kHz and 55.555 kHz MAS with the temperature of the cooling gas set
to 280 K and 235 K, respectively.
For 1D and 2D α-PET SH3 (H)N(H)H spectra, the ramped CP transfer from proton to nitrogen was
performed under the same conditions for all experiments: 42.95 kHz on the nitrogen channel and the optimal
ramped amplitude on the proton channel of 86.95-108.69 kHz. The mixing time was 1.05 ms. 9.3 kHz
WALTZ-16 (Shaka et al., 1983) with 25 us pulses and 10.4 kHz WALTZ-16 (Shaka et al., 1983) with 100
us pulses were applied on nitrogen and carbon channels during the acquisition. MISSISSIPPI water
suppression (Zhou and Rienstra, 2008) was applied for 100 ms with 13.513 kHz of the rf-field. The carrier
positions were set to 4.6 ppm, 118.5 ppm and 53.7 ppm for $^1$H, $^{15}$N and $^{13}$C, respectively, except where
otherwise indicated.
Table 1 summarizes the applied experimental parameters for 1D spectra.
**Table 1** Summary of the experimental parameters used in the 1D CP + RFDR (the start and the end values are shown) and HET-
RFDR using α-PET labeled SH3.

| | CP + RFDR | | HET-RFDR |
|---|---|---|---|
| | CP | RFDR | |
| $^1$H (kHz) | 86.95-108.69 | 86.21 | 86.21 |
| $^{15}$N (kHz) | 42.95 | - | 75.75 |
| transfer time (ms) | 0.55 | [0-7.776] | [0-7.776] |
| NS | 32 | | 32 |
| D1 (s) | 2 | | 2 |
| AQ (s) | 0.020448 | | 0.020448 |
| SW (kHz) | 25 | | 25 |

NS – number of scans; D1 – a recycle delay; AQ – the acquisition time; SW – the spectral width.



For 2D (H)N(H)H HET-RFDR spectra, during the indirect dimension 11.6 kHz SW$_f$-TPPM (Thakur et al.,
2006) decoupling with 36.36 us pulses was applied on the proton channel. Two mixing times were used:
1.152 ms and 3.456 ms. The widths of π-pulses on proton and nitrogen channels were 5.8 us and 6.6 us,
respectively. 16 scans were acquired per increment in $t_1$. The total time for the single 2D experiment was
10 hours. Table 2 summarizes the rest of the parameters.
**Table 2** Summary of the experimental parameters used in 2D HET HET-RFDR α-PET SH3 experiments.

|  | AQ1; AQ2 (s) | SW1;  SW2 (kHz) | DW1; DW2 (us) |
|---|---|---|---|
| HET-RFDR | 0.0527075;<br>0.020448 | 9.713;<br>25 | 102.94<br>20 |

1 and 2 are indirect and direct dimensions; AQ – the acquisition time; SW – the spectral width; DW – the dwell time.
For all 1D HC HET-RFDR experiments (Figure 3), 4.8 ms of the mixing time was applied. The widths of
π-pulses on proton and carbon channels were 5.8 us (86.21 kHz) and 6.6 us (75.75 kHz), respectively. 87
kHz SPINAL64 (Fung et al., 2000) with 6 us pulses was used during the acquisition. 128 scans were
accumulated. The spectral width was 50 kHz and the acquisition time 0.01536 s.
**Author Contributions**
EN performed the simulations and discovered heteronuclear HET-RFDR. EN and LBA designed
experiments. EN and KX recorded data. EN and LBA wrote the article. KTM prepared the SH3 protein
samples. All authors edited and approved the article.
**Competing Interests**
The authors declare that they have no conflict of interest.
**Acknowledgments**
We acknowledge financial support from the MPI for Biophysical Chemistry, and from the Deutsche
Forschungsgemeinschaft (Emmy Noether program Grant AN1316/1- 1)

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
