# Peer review of "Heteronuclear and Homonuclear Radio Frequency Driven"

_Magnetic Resonance, 2021_

## Author Comment (AC1)

We are extremely grateful to the reviewer for her/his time and effort in reviewing the manuscript.

The paper describes the use of rotor-synchronized RFDR pulse trains on two nuclei for simultaneous heteronuclear and homonuclear polarization transfer. Such a sequence can be used to replace a heteronuclear polarization-transfer step (e.g., CP) and a consecutive homonuclear polarization transfer step by a single element that achieves both transfers. The new method is demonstrated using simulations and experimental data on SH3. At fast MAS or probably more likely for rf pulses that occupy a large part of the rotor cycle, heteronuclear and homonuclear transfer are almost identical while for slow MAS or rf pulses that are short compared to the rotor cycle, little transfer is observed.

In Figure 2, the CP and RFDR overlay spectrum is virtually invisible. Please improve the graphic representation. It would also be more interesting to see a 1D trace comparison of the HET-RFDR and the CP+RFDR spectrum and not two spectra at different mixing times.

The updated Figure 2b now shows 2D HET-RFDR at 3.456 ms mixing with three 1D slices, where we compare CP+RFDR with HET-RFDR at two different mixing times (1.154 and 3.456 ms). Pages 5-6, lines 104-121 of the revised article now read:

*"Structurally interesting cross-peaks are indeed observed in the 2D HET-RFDR spectrum shown in Figure 2b at 3.456 ms mixing. For example, we have observed the amide-amide contact between V44 and V53, which is 4.82 Å in the crystal (pdb code 2NUZ (Castellani et al., 2002)). The amide to side chain contact of a A55 N to Hβ (3.41 Å) is also indicated in the Figure, along with a sequential contact from Y13 $^{15}$N to L12 $^{1}$Hα, which is 3.26 Å. These peaks are boxed in Figure 2b, and the 1D slices shown above the 2D spectra. For comparison, in 1D slices we show CP +RFDR (blue) and HET-RFDR (red) intensities of these three peaks for at two different mixing times: 1.154 ms (dashed) 3.456 ms (solid). Both methods provide similar intensities at long mixing time, whereas at shorter mixing times CP+RFDR provides higher intensities for short range distances*

[Figure]

***Figure 2*** *1D (a) and 2D (b) (H)N(H)H spectra of α-PET labeled SH3. For all spectra the first CP from proton to nitrogen was performed with 1.05 ms. (a) 1D spectra with different sequences used for the second transfer: CP + RFDR (blue) and HET-RFDR (red). For CP + RFDR, 0.55 ms of CP was used. For both RFDR and HET-RFDR, $t_{mix}$ of 0, 0.846, 1.728, 2.592, 3.456, 4.32, 5.184, 6.048, 6.912, 7.7776 ms are shown. (b) 2D HET-RFDR at 3.456 ms mixing. Spectra were recorded at a 600 MHz Bruker instrument equiped with a 1.3 mm probe and an MAS frequency of 55 kHz. The widths of π-pulses on proton and nitrogen channels were 5.8 us and 6.6 us, respectively. The 1D slices show the intensities of three selected peaks. CP+RFDR (blue) and HET-RFDR (red) at 1.154 and 3.456 ms mixing are displayed with dashed and solid lines , respectively. The experimental parameters are detailed in Table 1 and 2 the 'Experimental Methods'."*

I think it would be nice to discuss the different transfer characteristics of the two sequences since there are marked differences in cross peak appearance.

We added discussion about the different transfer characteristics of CP+RFDR and HET-RFDR signals. Page 5, lines 102-107 (the revised article):

*"Without RFDR mixing, the CP+RFDR detects directly bonded amide protons (Figure 2a, red with zero mixing time) and zero signal occurs for HET-RFDR (Figure 2a, blue) since the signal is on nitrogen. With increasing RFDR mixing, the signal is transferred from directly bonded amide protons to remote protons for the CP+RFDR sequence, whereas simultaneous transfer from nitrogen spins to amide protons and from amide protons to remote protons occurs with HET-RFDR."*

Why is the transfer to the Ca in Fig. 3f negative? Is this a double-quantum type Hamiltonian that is generated here?

We thank the reviewer for that valuable comment. We also observe the negative transfer polarization with simulations, but when at least three spin system is considered. For example, in Figure R1 we show the simulations under similar conditions as in Figure 3f. In all cases the initial operator was $I_z$ and the measured operator was $S_{z1}$. As can be seen, for two spin system the negative signal is very small (-0.005). However for three (red) and four (blue) spin systems the transfer polarizations achieve -0.05. It is not obvious via which operators the signal is transferred. Since the explanation of this negative peak is beyond the main goals of the article we decided not perform additional investigations. However, we added the simulation showing negative signal to SI.

[Figure]

**Fig. R1** The simulated HET-RFDR signals. The simulated HET-RFDR polarization transfers for *IS* (black) $IS_2$ (red) and $IS_3$ (blue) spin systems as a function of the mixing time. For all simulations MAS was 10 kHz and hard π-pulses with 5.8 us and 6.6 us widths were applied simuntaneously on *I* and $S_n$ spins every rotor period (XY8 phase cycling). The offset and dipolar coupling constants in kHz: *IS* - $[\Omega_I; \Omega_{S1}] = [0.5; 5.5]$, $[\nu_{D,IS1}] = [23]$; $IS_2$ - $[\Omega_I; \Omega_{S1}; \Omega_{S2}] = [0.5; 5.5; 7]$, $[\nu_{D,IS1}; \nu_{D,IS2}; \nu_{D,S1S2}] = [23; 3.4; 2.5]$; $IS_3 - [\Omega_I; \Omega_{S1}; \Omega_{S2}; \Omega_{S3}] = [0.5; 5.5; 7; -13]$, $[\nu_{D,IS1}; \nu_{D,IS2}; \nu_{D,IS3}; \nu_{D,S1S2}; \nu_{D,S1S3}] = [23; 3.4; 3.2; 2.5; 2.5]$. In all simulations the initial and the measured operators were $I_z$ and $S_{z1}$, respectively.

Can you please comment in the paper whether the observed variations in the transfer efficiency are in agreement with expectations from the simulations/operator analysis?

Yes, the observed transfer efficiency is in general agreement with expectations from simulations. Depending on values of the dipolar coupling constants, the simulated HET-RFDR transfer efficiency

achieves 20-35% for directly bonded spins (Figure S4 in SI). With respect to non-adiabatic CP (~70% transfer efficiency), the simulated HET-RFDR transfer efficiency is between 28-50%. In the experiment, we detected ~50% and ~40% of the CP signal in the HET-RFDR for directly bonded proton-carbon pairs (Figure S2b) and directly bonded proton-nitrogen pairs (Figure 2a), respectively.

Is there a reason why the rf-field amplitudes used in Figs. 4 and 5 are different? 83 vs 65 kHz? It would be much nicer to use the same rf field to make them directly comparable. I guess the simulation effort is not very high to change Fig. 5. I also think it would be nice to duplicate Fig. 5 in te SI for fast spinning so one can better understand the differences between the two cases in Fig. 4.

We have followed both suggestions, modifying Figure 5 accordingly. We added Figures 5a and b to the SI.

Castellani, F., van Rossum, B., Diehl, A., Schubert, M., Rehbein, K., and Oschkinat, H.: Structure of a protein determined by solid-state magic-angle-spinning NMR spectroscopy, Nature, 420, 99–102, https://doi.org/10.1038/nature01070, 2002.

---

## Author Comment (AC2)

We are extremely grateful to the reviewer for her/his positive comments, the time and effort in reviewing the manuscript.

This is an interesting and potentially important manuscript, demonstrating the utility of simultaneous RFDR pi pulse trains on 15N and 1H as a means of transferring polarization from 15N to 1H and also transferring polarization among 1H spins. The authors show that long-range 1H-1H transfers are observed, which provide useful structural information. With numerical simulations, they explore dependences on resonance offsets that are important especially at lower MAS frequenices.

This paper is certainly suitable for publication in MR. My only recommendation is that the authors re-examine their choice of references in the Introduction. It is worth noting that the first examples of homonuclear dipolar recoupling (by Meier and Earl for 1H-1H couplings and by Tycko and Dabbagh for 13C-13C couplings and quantitative distance measurements) are not cited.

We apologize for the missing articles. We included the articles of Meier and Earl, (1986) and Tycko and Dabbagh, (1990) into citation.

The Ok paper is not about RFDR.

We corrected the wrong citation on line 41, page 2. It should be the article of Bennett et al., (1992).

What is now called RFDR was originally introduced by Gullion and Vega (and called SEDRA). The 2001 paper by Ishii analyzed finite-pulse effects in detail and showed that XY4 phase cycling produces an average homonuclear dipole-dipole Hamiltonian with the same operator form as in a non-spinning sample.

We added the following sentence and the citation about the article of Gullion and Vega:

*"The recoupling of the homonuclear dipolar interactions with a train of $\pi$-pulses every rotor period was originally introduced by Gullion and Vega (Gullion and Vega, 1992) and Bennett et all (Bennett et al., 1992b)."*

Bennett, A. E., Griffin, R. G., Ok, J. H., and Vega, S.: Chemical shift correlation spectroscopy in rotating solids: Radio frequency-driven dipolar recoupling and longitudinal exchange, J. Chem. Phys., 96, 8624–8627, https://doi.org/10.1063/1.462267, 1992.

Gullion, T. and Vega, S.: A simple magic angle spinning NMR experiment for the dephasing of rotational echoes of dipolar coupled homonuclear spin pairs, Chem. Phys. Lett., 194, 423–428, https://doi.org/10.1016/0009-2614(92)86076-T, 1992.

Meier, B. H. and Earl, W. L.: Excitation of multiple quantum transitions under magic angle spinning conditions: Adamantane, J. Chem. Phys., 85, 4905–4911, https://doi.org/10.1063/1.451726, 1986.

Tycko, R. and Dabbagh, G.: Measurement of nuclear magnetic dipole—dipole couplings in magic angle spinning NMR, Chem. Phys. Lett., 173, 461–465, https://doi.org/10.1016/0009-2614(90)87235-J, 1990.